# The Combined Effects on Human Dental Pulp Stem Cells of Fast-Set or Premixed Hydraulic Calcium Silicate Cements and Secretome Regarding Biocompatibility and Osteogenic Differentiation

**DOI:** 10.3390/ma17020305

**Published:** 2024-01-07

**Authors:** Yun-Jae Ha, Donghee Lee, Sin-Young Kim

**Affiliations:** 1Department of Conservative Dentistry, Seoul St. Mary’s Hospital, College of Medicine, The Catholic University of Korea, Seoul 06591, Republic of Korea; gkdbswo1122@naver.com; 2Department of Dentistry, College of Medicine, The Catholic University of Korea, Seoul 06591, Republic of Korea; dong524@catholic.ac.kr

**Keywords:** cell migration, cell viability, hydraulic calcium silicate cement, osteogenic ability, secretome

## Abstract

An important part of regenerative endodontic procedures involving immature permanent teeth is the regeneration of the pulp–dentin complex with continuous root development. Hydraulic calcium silicate cements (HCSCs) are introduced for the pulpal treatment of immature permanent teeth. The stem-cell-derived secretome recently has been applied for the treatment of various damaged tissues. Here, we evaluated the biocompatibility and osteogenic differentiation of HCSCs combined with secretome on human dental pulp stem cells. In the Cell Counting Kit-8 test and wound healing assays, significantly higher cell viability was observed with secretome application. In alkaline phosphatase analysis, the activity was significantly higher with secretome application in all groups, except for RetroMTA on day 2 and Endocem MTA Premixed on day 4. In an Alizarin Red S staining analysis, all groups with secretome application had significantly higher staining values. Quantitative real-time polymerase chain reaction results showed that the day 7 expression of *OSX* significantly increased with secretome application in all groups. *SMAD1* and *DSPP* expression also increased significantly with secretome addition in all groups except for Biodentine. In conclusion, HCSCs showed favorable biocompatibility and osteogenic ability and are predicted to demonstrate greater synergy with the addition of secretome during regenerative endodontic procedures involving immature permanent teeth.

## 1. Introduction

With the development of tissue engineering, the regeneration and replacement of new tissues using stem cells, growth factors, and scaffolds have been studied in the field of regenerative medicine. Within endodontics, regenerative endodontics is emerging as an area of focus for addressing necrotic pulp tissue. The aim of a regenerative endodontic procedure is to replace damaged tooth structures, including the root, dentin, and cells within the pulp–dentin complex [1]. Rehabilitation of the pulp–dentin complex takes on greater significance when immature permanent teeth suffer from necrosis.

Mineral trioxide aggregate (MTA) is applied extensively in pulp therapy for immature permanent teeth, offering exceptional biocompatibility, antibacterial properties, and an impressive sealing capability [2]. MTA is a biomaterial and has been used in various endodontic treatments, such as vital pulp therapy, regenerative endodontic procedures, repair of perforation sites, and retrograde filling during endodontic microsurgery [3,4,5,6,7]. The classic MTA, ProRoot MTA (Dentsply Tulsa Dental Specialties, Tulsa, OK, USA), is renowned for its capacity to enhance osteogenic and dentinogenic markers as well as promote angiogenesis in human dental pulp stem cells (hDPSCs) [3]. Its disadvantages are related to its long setting time, the likelihood of discoloration, and its heavy metal content [4,5,6,7].

To compensate for these shortcomings, various MTAs and hydraulic calcium silicate cements (HCSCs) have been developed [8,9]. HCSCs were intended to shorten the setting time and overcome discoloration [10,11,12]. RetroMTA (BioMTA, Seoul, Republic of Korea) has biocompatibility and physical properties that are similar to those of ProRoot MTA, and its initial setting time is faster than ProRoot MTA, at 150 s [13,14]. Among HCSCs, in animal and clinical trials, Biodentine (Septodont, Saint-Maur-dens-Fossés, France) has shown bone-formation and dentin-formation capabilities similar to those of ProRoot MTA, without cytotoxicity, and its initial setting time is also faster than ProRoot MTA with 12 min [15,16]. Additionally, a newly developed material, Endocem MTA Premixed (Maruchi, Wonju, Republic of Korea), is an injectable premixed material that offers technicians a simplified clinical option. [17,18]

The secretome derived from human dental stem cells, including stem cells derived from human exfoliated deciduous teeth (SHEDs), has emerged as a promising candidate for addressing various medical conditions. It shows potential in mitigating neurodegenerative diseases, repairing nerve damage, treating cartilage defects, and promoting bone regeneration through mechanisms that include nerve protection, anti-inflammatory effects, cell death inhibition, and the stimulation of angiogenesis [19]. As indicated recently, the secretome derived from hDPSCs demonstrates a versatile applicability in the field of tissue regeneration. This versatility is attributed to its multifaceted paracrine effects, and notably, it exhibits a remarkable capacity for osteogenesis [19,20].

The objective of this study was to evaluate and compare the biocompatibility and osteogenic differentiation capacity of ProRoot MTA, Biodentine, RetroMTA, and Endocem MTA Premixed when used in combination with secretome on hDPSCs.

## 2. Materials and Methods

### 2.1. Human Dental Pulp Stem Cells (hDPSCs)

This study received approval from the Institutional Review Board of Seoul St. Mary’s Hospital, the Catholic University of Korea (IRB No. MC22ZASI0063). We employed hDPSCs that are commercially available from Top Cell Bio Inc. (Seoul, Republic of Korea) (SH0010, Passage 4). The company’s evaluation confirmed that CD29, CD44, CD73, and CD105 expression was >90% positive in all samples of these cell lines.

Cells were cultivated in a growth medium, which included HyClone minimum essential medium (α-MEM; Cytiva, Marlborough, MA, USA), supplemented with 10% fetal bovine serum (Cytiva), penicillin, and streptomycin. To assess osteogenic potential, we used an osteogenic medium consisting of α-MEM along with beta-glycerophosphate (Sigma-Aldrich, St. Louis, MO, USA), dexamethasone (Sigma-Aldrich), and ascorbic acid (Sigma-Aldrich). Cell cultures were incubated in a controlled environment at 37 °C under 5% CO_2_, with rigorous adherence to sterile conditions throughout all experimental procedures.

### 2.2. Disks of Different MTAs and HCSCs for Experimentation

ProRoot MTA and HCSCs (RetroMTA, Biodentine, Endocem MTA Premixed), and the SHEDs-derived secretome (Top Cell Bio Inc.) were used in this study. Table 1 lists the composition details for each of them. With strict adherence to sterile protocols, we prepared each cement following the manufacturer’s guidelines and produced specimens in the form of disks measuring 6 mm in diameter and 3 mm in height. ProRoot MTA powder was mixed with the supplied liquid in a ratio of 3:1, one capsule of Biodentine powder was mixed with five drops of supplied liquid using a mixer, one cap of RetroMTA powder was mixed with three drops of supplied liquid, and Endocem MTA Premixed was directly applied to the mold. Subsequently, all samples were enveloped in wet gauze and left to harden for 72 h within a room-temperature clean bench. Afterward, they underwent a 4 h sterilization process using ultraviolet rays within a clean bench.

### 2.3. Categorization of Experimental Groups

Experimental groups were developed as follows: a control group without secretome, consisting of hDPSCs cultured without experimental disks; a control group with secretome, consisting of hDPSCs cultured without experimental disks, and with culture media supplemented only with secretome; ProRoot MTA without secretome; ProRoot MTA with secretome; Biodentine without secretome; Biodentine with secretome; RetroMTA without secretome; RetroMTA with secretome; Endocem MTA Premixed without secretome; and Endocem MTA Premixed with secretome.

### 2.4. Cell Viability Assay

The cytotoxic effects of MTAs and HCSCs were evaluated using a Cell Counting Kit-8 test (CCK-8) (CK04-13; Dojindo, Kumamoto, Japan). The cell proliferation rate of hDPSCs was determined at different timepoints, including immediately after culturing (0), as well as at 2, 4, and 6 days post-culture. hDPSCs were plated into 24-well cell culture plates (SPL Life Sciences, Pocheon, Republic of Korea) at a density of 2.5 × 10^4^ cells per well in a growth medium. After 24 h of cell adhesion culture, absorbance immediately after culture was measured.

Each disk was positioned on a cell co-culture platform (SPLInsert; SPL Life Sciences) featuring 0.4 µm pores, and this platform was positioned directly above the hDPSCs in each well. To ensure that the culture solution remained at the full height of the disk, we supplemented each well with an extra 1 mL of culture solution. Control groups consisted of hDPSCs cultured without experimental disks. We introduced 20 µL of CCK-8 solution into each well, followed by incubation at 37 °C for 1 h. We then measured the absorbance at 450 nm, using 650 nm as the reference point, with the aid of an absorbance microplate reader (PowerWave XS; BioTek Instruments, Winooski, VT, USA). Eight independent samples were assessed for each group.

### 2.5. Cell Migration Assay

We assessed cell migration ability using a wound healing assay. hDPSCs were cultured in a 24-well cell culture plate at a density of 3.5 × 10^4^ cells per well in a growth medium. Following a 24 h incubation period, using a 1000 µL pipette tip, we created a scratch in the center of the cell layer adhered to the well floor. After the scratching, isolated cell substances were washed with phosphate-buffered saline. After 24 h of incubation, each disk was placed on a SPLInsert with pores of 0.4 µm, and the platform was placed above hDPSCs in each well. To maintain the culture solution to the full height of the disk, each well was supplemented with an additional 1 mL of culture solution. hDPSCs with experimental disks were cultured for 4 days, and we obtained wound healing images at 0, 1, 2, 3, and 4 days using a phase-contrast microscope from Olympus (Tokyo, Japan). The area covered by cell migration towards the scratch site was quantified using ImageJ 1.46r software (National Institutes of Health, Bethesda, MD, USA). Cell migration into the scratch area was then calculated as a percentage relative to the initial scratch size. Four independent samples were assessed for each group.

### 2.6. Alkaline Phosphatase (ALP) Activity

To evaluate the osteogenic potential of hDPSCs, we conducted ALP analysis at 2 and 4 days. Cells were cultivated at a density of 0.7 × 10^4^ cells per well in a 24-well cell culture plate containing osteogenic medium. Individual disks were positioned on an SPLInsert with 0.4 µm pores, situated above the hDPSCs in each well. To maintain the culture solution to the full height of the disk, we supplemented each well with an additional 1 mL of culture solution. On days 2 and 4, each well was rinsed with phosphate-buffered saline, followed by the addition of 20 µL of a dissolution buffer (0.2% Triton X-100; AnaSpec, Fremont, CA, USA). The blend was incubated at 37 °C for 15 min. Subsequently, we detached the adherent cells by scraping, transferred the solution to a 1.5 mL microcentrifuge tube, and centrifuged it at 4 °C for 10 min at 2500× *g*. The resulting supernatant was collected for ALP analysis. For the ALP analysis, 50 µL of p-nitrophenyl phosphate (pNPP) ALP substrate solution (AnaSpec, Fremont, CA, USA) was added to each well. pNPP was placed on top of each sample, and the reagent was gently mixed for 30 s. After incubation at 4 °C for 30 min, absorbance was measured at 405 nm with a PowerWave XS. Six independent samples were assessed for each group.

### 2.7. Alizarin Red S (ARS) Staining Assay

We used the ARS staining assay to assess calcium deposition within hDPSCs. Each experimental disk was combined with an osteogenic medium and stored in a 100% humidity incubator at 37 °C for 6 days, resulting in a concentration of 5 mg/mL. hDPSCs were cultured in a 24-well plate at a density of 0.7 × 10^4^ cells per well and exposed to an experimental disk exudate solution for 12 days. Subsequently, we fixed the cells using a 4% paraformaldehyde solution and a 2% ARS solution from ScienCell (Carlsbad, CA, USA) for 20 min. The dyeing process involved treatment with 10% cetylpyridinium chloride (Sigma-Aldrich) for 15 min, followed by absorbance measurement at 560 nm using a PowerWave XS. Six independent samples were assessed for each group.

### 2.8. Quantitative Real-Time Polymerase Chain Reaction (qRT-PCR)

Seven days after culture, we obtained total RNA from each cell with the RNeasy Mini Kit (Qiagen, Hilden, Germany). Subsequently, we initiated RNA reverse transcription reactions using the RevertAid First Strand cDNA Synthesis Kit (Thermo Fisher Scientific, Waltham, MA, USA) to synthesize cDNA. For the analysis, we employed primers for key markers, including Runt-related transcription factor 2 (*RUNX2*), osterix (*OSX*), and Suppressor of Mothers against Decapentaplegic (*SMAD1*), which are known osteogenic markers, as well as dentin sialophosphoprotein (*DSPP*), recognized as a dentinogenic marker. Primers were designed using GenBank (Table 2).

We isolated the entire RNA from hDPSCs using Tri Reagent (TR118; Molecular Research Center, Inc., Cincinnati, OH, USA). Following quantification of the total RNA concentration, we used 500 ng as a template. For qRT-PCR analysis, we employed the iTaq Universal SYBR Green One-Step Kit (Bio-Rad, Hercules, CA, USA) in conjunction with the CFX96 Real-Time PCR Detection System (Bio-Rad), and the data were analyzed with Bio-Rad CFX Manager software, version 3.1. The denaturation and annealing processes in qRT-PCR were as follows: 45 cycles, beginning with an initial step at 50 °C for 10 min, followed by an initial denaturation at 95 °C for 1 min. Subsequently, denaturation continued at 95 °C for 10 s, succeeded by an annealing step encompassing 60 cycles at 60 °C for 30 s, with a final extension at 60 °C for 5 s. For the melting curve analysis, the temperature was incrementally increased by 0.5 °C per cycle. The expression levels of amplified genes were calculated relative to that of *GAPDH*, and all gene expression was compared with the control group. Three independent samples were assessed for each group.

### 2.9. Statistical Analysis

Statistical analysis was performed with the SPSS software program (version 24.0; IBM Corp., Armonk, NY, USA). The distribution of data was assessed for normality using the Shapiro–Wilk test. To assess significant differences among time and group factors for each experimental method, we used repeated measures analysis of variance (ANOVA). After confirmation of group factor significance, a one-way ANOVA was conducted, followed by Tukey post hoc test. To assess the impact of applying Secretome, an independent *t*-test was employed. A statistically significant threshold was set at *p* < 0.05.

## 3. Results

### 3.1. Cell Viability Assay

CCK-8 measurements were taken on days 2, 4, and 6 and indicated a steady increase in cell proliferation over time (Figure 1). Of note, on all 3 days, the experimental groups supplemented with secretome exhibited significantly higher cell viability compared to their counterparts without secretome (Figure 1, *p* < 0.05). On day 2, the Biodentine and Endocem MTA Premixed groups, both without secretome, displayed significantly higher cell viability compared to the control group without secretome (Figure 1a, *p* < 0.05). However, there were no significant differences between each group without secretome on days 4 and 6 (Figure 1b,c).

### 3.2. Cell Migration Assay

The cell migration rate increased significantly over time regardless of the presence or absence of secretome, without group-specific effects (Figure 2). On days 1 and 2, the cell migration rate in all experimental groups with secretome except the control group was significantly higher than in experimental groups without secretome (Figure 2a,b, *p* < 0.05). On day 3, there was no significant difference according to the presence or absence of scretome in each experimental group (Figure 2c, *p* > 0.05). Representative images for each group are shown in Figure 3.

### 3.3. ALP Activity

ALP activity was measured on days 2 and 4, and higher ALP activity was observed over time (Figure 4). On day 2, ALP activity was significantly higher in all groups with secretome application compared to the corresponding groups without secretome, except for the RetroMTA group (Figure 4a, *p* < 0.05). On day 4, ALP activity was significantly higher in all groups with versus without secretome except for the Endocem MTA Premixed group (Figure 4b, *p* < 0.05). On day 2, the results were significantly higher in the RetroMTA group without added secretome compared with the other groups (Figure 4a, *p* < 0.05).

### 3.4. ARS Staining Assay

ARS staining was performed on days 6 and 12 after incubation, and ARS staining levels were observed to increase over time (Figure 5). On day 6, the ARS staining level in the RetroMTA group was higher than in other groups with or without secretome. On day 6, for all experimental groups and the control group, values were significantly higher with versus without secretome (Figure 5a, *p* < 0.05). On day 12, values were significantly higher in all experimental groups than in the control group, regardless of whether secretome was applied (Figure 5b, *p* < 0.05), and with versus without secretome in all groups, including the control group (Figure 5b, *p* < 0.05). The ARS staining images for each group are shown in Figure 6. From day 6 to day 12, the area of calcium accumulation increased, and, with secretome application, the ARS staining area increased.

### 3.5. qRT-PCR

With the application of secretome, the expression level of all markers was generally increased (Figure 7). This increase was especially prominent for *OSX* in all experimental groups with secretome compared to their respective groups without it (Figure 7b, *p* < 0.05). In the case of *SMAD1*, the Endocem MTA Premixed group showed increased expression compared to other groups regardless of secretome application (Figure 7c, *p* < 0.05). Both *SMAD1* and *DSPP* expression levels increased with secretome in all but the Biodentine group compared to the respective groups without secretome (Figure 7c,d, *p* < 0.05).

## 4. Discussion

When the pulp of an immature permanent tooth with a wide root apex shows necrosis, calcium hydroxide has historically been used to form hard tissue at the root apex. However, if calcium hydroxide is used inside the root canal for a long period, the root canal wall can weaken, leading to root fracture, with the disadvantage of this requiring a long treatment period [1]. To compensate, MTA, with its excellent sealing power, biocompatibility, and antibacterial properties, was introduced for the pulp therapy of immature permanent teeth [2]. However, because a regenerative endodontic procedure for immature permanent teeth is carried out in a blood environment, and the hardening of the material can be reduced, HCSCs have been developed to improve on the disadvantages of existing MTAs, such their as different operability and long hardening time [21,22].

Recently, a secretome derived from hDPSCs has been applied in the treatment of various damaged tissues and has shown significant osteogenic potential because of its various paracrine effects [19]. Mesenchymal stem cells derived from hDPSCs are emerging as a promising tool, with great potential for tooth and oral tissue regeneration, and have been used in clinical trials as a tool in regenerative medicine [23].

In this study, given the osteogenic potential and tissue regeneration effects of secretome, we sought to determine if its use in addition to MTA or HCSCs could increase the success rate of vital pulp therapy or regenerative endodontic procedures. For this purpose, we assessed the biocompatibility and osteogenic potential of ProRoot MTA, Biodentine, RetroMTA, and Endocem MTA Premixed, each with added secretome.

The key finding of this study was that all experimental hydraulic calcium silicate cements showed favorable biocompatibility like ProRoot MTA, and significantly higher cell viability was observed with secretome application. Regarding osteogenic potential, all experimental hydraulic calcium silicate cements showed higher mineralization activity when secretome was combined, as did ProRoot MTA.

To assess biocompatibility, we used CCK-8 analysis and the wound healing assay. CCK-8 relies on Dojindo’s water-soluble tetrazolium salt (WST-8) in a sensitive colorimetric assay to determine viable cell counts. WST-8 is reduced by cellular dehydrogenase to produce formazan, which dissolves in tissue culture medium, and CCK-8 leverages the fact that the amount of formazan dye is directly proportional to the number of living cells [24]. The methyl thiazol tetrazolium (MTT) assay also is widely used to evaluate cell proliferation and cytotoxicity [25] but has the disadvantage of underestimating cell damage and detecting cell death only in the late stages when the cell metabolism is significantly reduced [26]. Therefore, in this study, we used the CCK-8 assay, which has superior sensitivity and low cytotoxicity compared with the MTT assay and allows for repeated measurements of the same sample. The wound healing assay is used to evaluate biocompatibility by observing cell migration to the scratch site [27].

CCK-8 and wound healing assay results showed an overall trend of cell proliferation over time. Based on CCK-8 assay findings, cells proliferated better when secretome was applied, even on day 6. In the wound healing assay, however, secretome had an effect only in the early stages, and cell migration was not faster when secretome was added from day 3. MTA and HCSCs did not have a negative effect on cell survival and migration and showed a higher cell viability than the control group on day 2 based on the CCK-8 assay. A previously published study also showed that when MTA was applied to SHEDs, cell viability was similar to or higher than that of the control group [28]. In addition, other studies have shown that tricalcium silicate, an ingredient in ProRoot MTA and Biodentine, induces the proliferation of human dental pulp cells [29], and that RetroMTA, which contains calcium carbonate, yielded better cell viability compared with a control group [30]. These previous results are consistent with the current findings. Moreover, the additional application of secretome appears to further increase biocompatibility, similar to previous results showing that the conditioned medium of hDPSCs significantly increases cell differentiation and migration [31].

In regenerative endodontic procedures on immature permanent teeth, the increased root wall is reported to be formed by the growth of cementum, bone, or dentin-like tissue, based on histological observation. In a study using teeth extracted after a regenerative endodontic procedure, newly formed mineralized tissue along the dentin wall was cementum-like tissue or osteoid dentin [32]. In the current work, secretome was mixed with the expectation that it would affect regeneration rather than repair when applied with MTA and HCSCs to hDPSCs. We used ALP activity, ARS analysis, and qRT-PCR to evaluate the osteogenic and dentinogenic potential of these combinations.

ALP initially attaches to the outside of the cell membrane, including in osteoblasts, and is used as a marker to distinguish cells. When active bone formation increases, the expression of ALP increases, which means that osteoblast differentiation actively occurs [33]. We found significantly higher ALP activity in the MTA and HCSC groups compared with the control group at day 4, and when secretome was applied during the entire experimental period, ALP activity significantly increased in the ProRoot MTA and Biodentine groups. Thus, MTA or HCSCs support hDPSC’s differentiation into osteoblasts, and the effect appears to be greater when secretome is applied. Silva et al. reported that MTA exhibits a high pH, which can neutralize acids secreted by osteoclasts and help prevent the further destruction of mineralized tissue [34]. In one study, ProRoot MTA in interaction with water generated calcium hydroxide and calcium ion release, leading to a high pH and mineralization, and consequently increased ALP activation [35]. In addition, previous studies have reported that Biodentine, RetroMTA, and Endocem MTA Premixed increase ALP activation [17,18], which supports the current finding that the experimental groups treated with HCSCs on day 4 had higher ALP values than the control group.

The ARS staining assay is used to determine the degree of osteogenic differentiation by evaluating calcium accumulation in cell culture. Here, we observed that ARS staining values increased on days 6 and 12 in all groups with secretome compared to the respective groups without secretome. On day 12, ARS staining values for the experimental groups with MTA or HCSCs were higher than the control values, regardless of whether secretome was applied. MTA and HCSCs thus increase calcium accumulation in cells, an effect that is further increased when secretome is added. Previous studies have shown that secretome derived from human gingiva demonstrates osteogenic potential in the form of significant calcium deposits, even after 6 weeks [20]. This finding is similar to the current results, in which calcium nodule formation continued to amplify for up to 12 days when secretome was added.

In this study, qRT-PCR was used to analyze the expression levels of the osteogenic markers *RUNX2*, *OSX*, and *SMAD1* and the dentinogenic marker *DSPP* in each experimental group. *RUNX2* and *OSX* are early markers of osteoblast differentiation [36,37], and *RUNX2* is expressed in predontoblasts, is essential for tooth formation, and is involved in the mineralization of tooth tissue [38,39]. SMAD1 is a protein involved in enhancing the differentiation of hDPSCs into odontoblasts and osteoblasts [40]. DMP-1 is a non-collagenous protein that is important for the mineralization of both bone and tooth [41]. There are reports that *DMP-1* is observed in abundance in the early stage of odontoblast differentiation, and *DSPP* is reported to be observed in the predentin layer but was also found in fully polarized dentin in the late stage of differentiation [42]. In this study, when secretome was not applied, overall, there was no statistically significant difference at 7 days in *RUNX2* and *OSX* expression levels between the control group and all experimental groups. Thus, it is not possible to confirm from these findings that adding MTA or HCSCs amplified the expression of osteogenic and dentinogenic genes compared to the control group. However, when secretome was applied, there were generally significantly highly expressed gene levels in the control group, as well as in all experimental groups. In the case of the ProRoot MTA, RetroMTA, and Endocem MTA Premixed groups, *OSX*, *SMAD1*, and *DSPP* showed higher expression levels when secretome was added. Applying secretome, therefore, can enhance osteogenic and dentinogenic cell differentiation of hDPSCs, and adding secretome to MTA or HCSCs can further increase this effect. In a previous study, ProRoot MTA application led to the increased expression of *osteocalcin (OCN*), *SMAD1*, *OSX*, *DMP-1*, and *DSPP* compared with controls, and the expression of *OCN*, *SMAD1*, *OSX*, and *DSPP* was amplified with the application of Endocem MTA Premixed [18]. Another study showed that *RUNX2* expression increased with the addition of Biodentine [43,44], consistent with the current results.

The main purpose of pulp therapy for immature permanent teeth is to preserve the activity and function of the remaining pulp tissue and to increase root wall thickness and root length. Accordingly, the materials used for the pulp therapy of immature permanent teeth must induce the differentiation of hDPSCs into osteogenic and dentinogenic cells to enable the formation of the pulp–dentin complex and reparative dentin [45]. Here, we found that MTA and HCSCs increased osteogenic potential, and when secretome was added, both osteogenic and dentinogenic potential improved. In this way, using MTA and HCSCs together with secretome in the pulp therapy of immature permanent teeth could more effectively achieve root wall thickness and root length growth.

Recently, increasing evidence indicates the importance of paracrine signaling induced by mesenchymal stem cells as a support mechanism for the regeneration of damaged tissues [46,47]. Although some studies have shown that the differentiation capacity of mesenchymal stem cells is not the primary mechanism for repairing damaged tissue in most diseases, tissue repair properties may be attributed to bioactive factors secreted by these stem cells that contribute to their paracrine activity [46,47,48,49]. Cell-free treatment using the secretome derived from mesenchymal stem cells could represent a new approach in regenerative medicine [46,47,49,50,51]. Secretome derived from hDPSCs and SHEDs may contribute to promoting new bone formation, tooth tissue regeneration, and nerve regeneration through the secretion of proangiogenic factors [19,20], and promote bone formation by enhancing the expression of osteogenic genes including *RUNX2*, *OCN*, *osteopontin*, and *OSX* [19], in keeping with the current results. In vitro studies have shown that secretome derived from DPSCs can have a stimulating effect on odontoblast differentiation, and in vivo studies point to its induction of the regeneration of pulp-like tissue [20].

This study had several limitations. It was conducted on cells formed as a monolayer in vitro, and in vivo research is needed. The problem of allogenic immune responses in major histocompatibility complex-mismatched recipients reported in previous studies must be considered when applying the secretome in vivo. In addition to this experiment, further methods to evaluate physical and chemical properties (e.g., solubility, volume stability, pH, Vickers hardness test) should be implemented to compare the differences and superiority between materials. There could be potential discrepancies between the material composition provided by the manufacturer and its actual surface composition. Despite these limitations, a conclusion can be drawn based on the current findings. The various HCSCs developed to overcome the shortcomings of existing MTAs in vital pulp therapy or regenerative endodontic procedures on immature permanent teeth show good biocompatibility and osteogenic potential, in agreement with other available evidence. If secretome is added, a more significant synergistic effect on cell viability and osteogenic potential can be expected.

## Figures and Tables

**Figure 1 materials-17-00305-f001:**
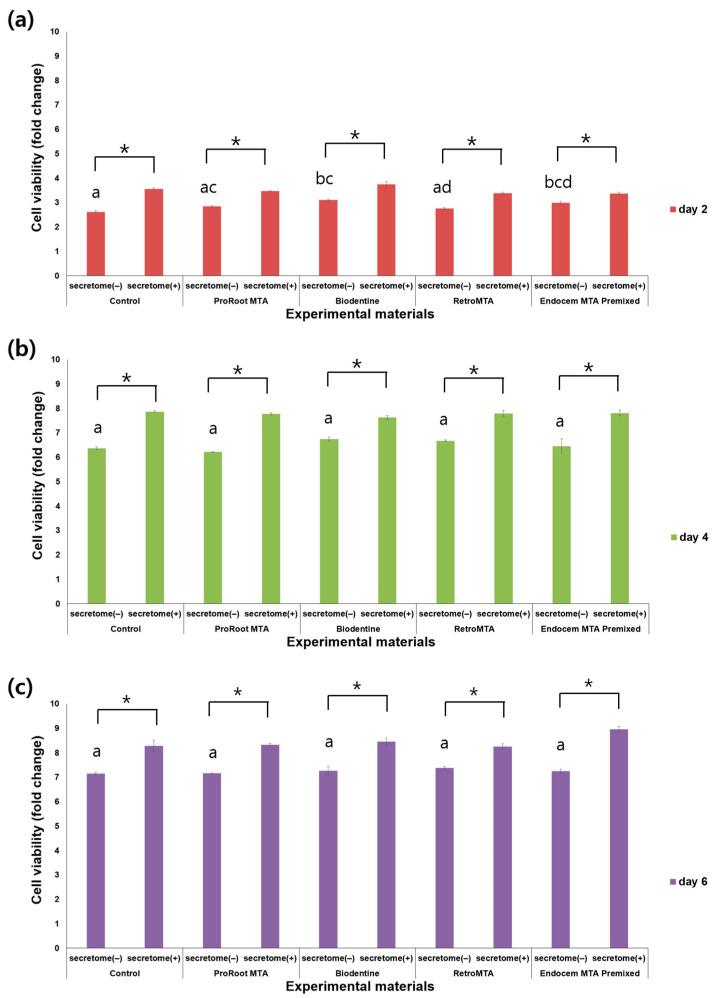
Comparison of cell viability values by material with or without secretome at (**a**) day 2, (**b**) day 4, and (**c**) day 6. Different characters indicate statistical significance between compared groups, and * denotes statistical significance within the indicated comparison.

**Figure 2 materials-17-00305-f002:**
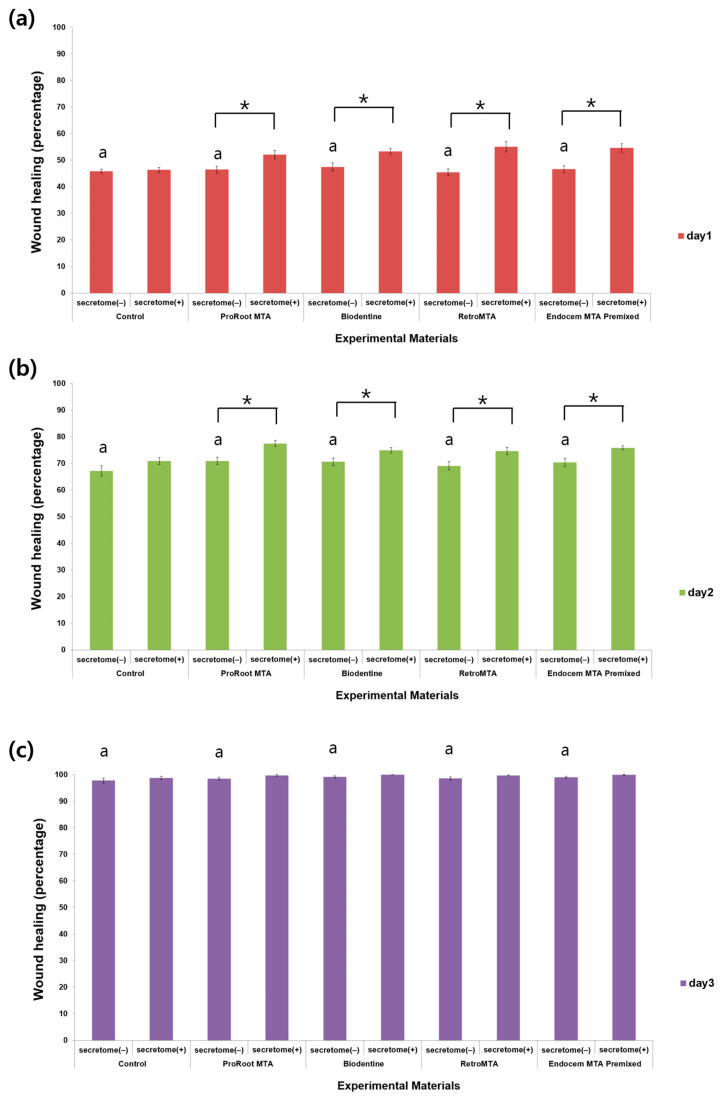
Comparison of the cell migration values for each material with or without secretome on (**a**) day 1, (**b**) day 2, and (**c**) day 3 of the wound healing assay. Different characters indicate statistical significance between compared groups, and * denotes statistical significance within the indicated comparison.

**Figure 3 materials-17-00305-f003:**
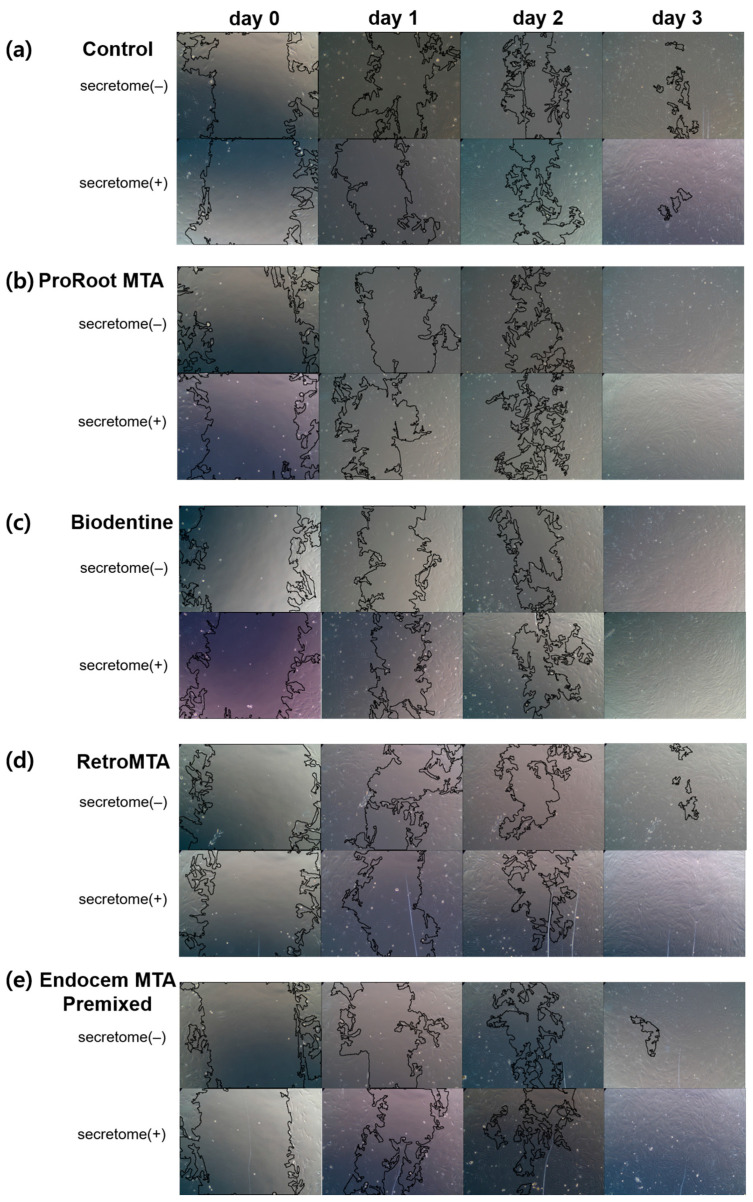
Comparison of representative images from wound healing assay. (**a**) The control group, (**b**) the ProRoot MTA group, (**c**) the Biodentine group, (**d**) the RetroMTA group, (**e**) the Endocem MTA Premixed group.

**Figure 4 materials-17-00305-f004:**
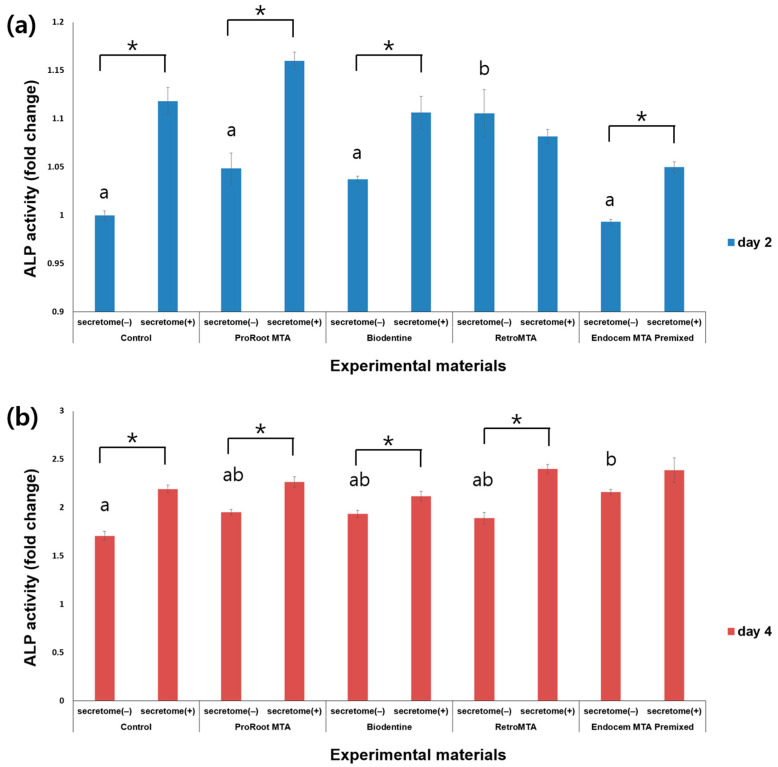
Comparison of ALP activity for each material according to the presence or absence of secretome on (**a**) day 2 and (**b**) day 4. Different characters indicate statistical significance between compared groups, and * denotes statistical significance within the indicated comparison.

**Figure 5 materials-17-00305-f005:**
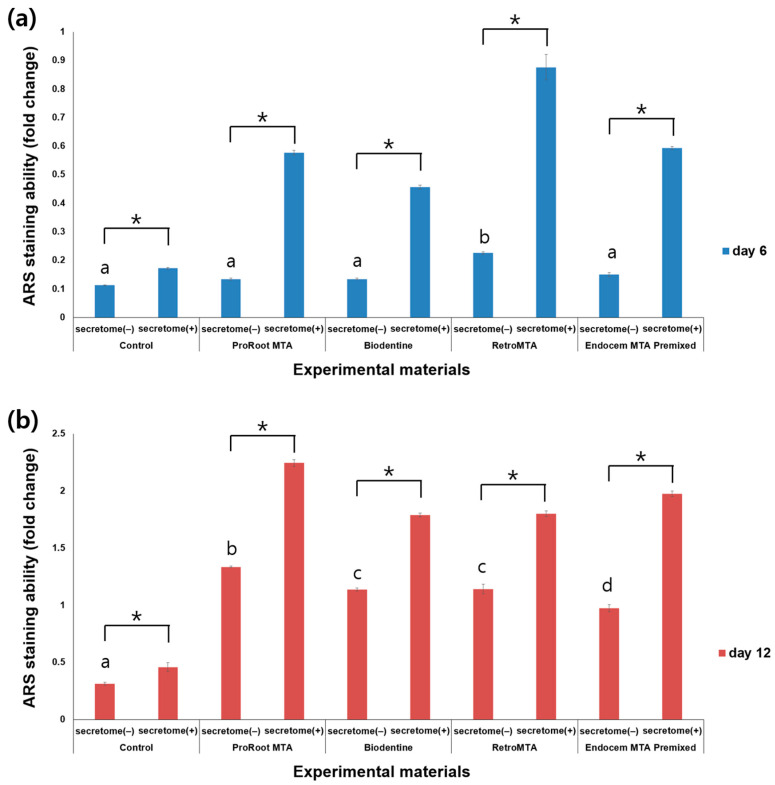
Comparison of ARS staining for each material with or without secretome on (**a**) day 6 and (**b**) day 12. Different characters indicate statistical significance between compared groups, and * denotes statistical significance within the indicated comparison.

**Figure 6 materials-17-00305-f006:**
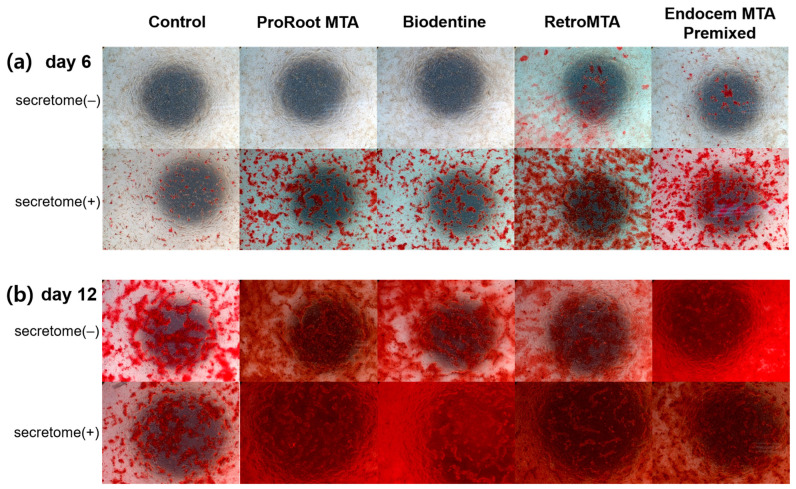
Representative images from the ARS staining assay on (**a**) day 6 and (**b**) day 12.

**Figure 7 materials-17-00305-f007:**
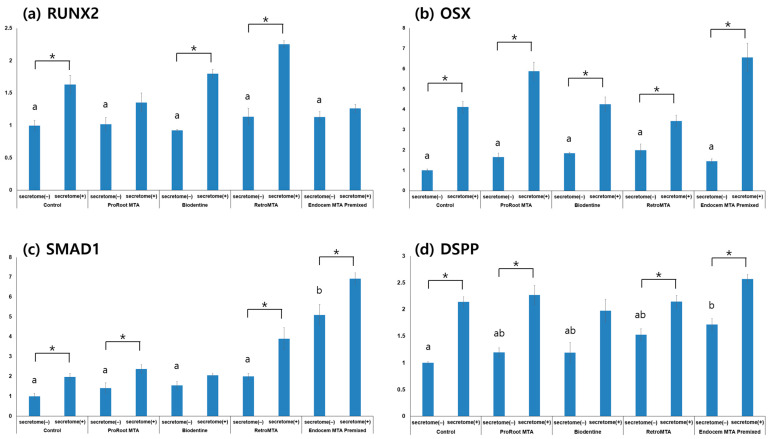
Comparison of qRT-PCR results in each experimental group on day 7. (**a**) *RUNX2*, (**b**) *OSX*, (**c**) *SMAD-1*, (**d**) *DSPP*. Different characters indicate statistical significance between compared groups, and * denotes statistical significance within the indicated comparison.

**Table 1 materials-17-00305-t001:** Materials: manufacturers and chemical compositions.

Material	Manufacturer	Composition	Powder Size	Batch Number
ProRoot MTA	Dentsply Tulsa Dental Specialties, Tulsa, OK, USA	Portland cement (tricalcium silicate, dicalcium silicate, and tricalcium aluminate) 75% Calcium sulfate dihydrate (gypsum) 5% Bismuth oxide 20%	6.9 μm	294002
Biodentine	Septodont, Saint-Maur-dens-Fossés, France	Tricalcium silicate 80.1% Calcium carbonate 14.9% Zirconium Oxide 5% Calcium chloride and polycarboxylate as an aqueous liquid	3.77 μm	B29557
RetroMTA	BioMTA, Seoul, Republic of Korea	Calcium carbonate 60–80% Silicon dioxide 5–15% Aluminum oxide 5–10% Calcium zirconia complex 20–30%	2.62 μm	RMCA04D03
Endocem MTA Premixed	Maruchi, Wonju, Republic of Korea	Zirconium dioxide 45–55% Calcium silicate 20–25% Calcium aluminate 1–5% Calcium sulfate 1–5% Dimethyl sulfoxide 20–25% Thickening agent 1–5%		FD220323A
secretome	Top Cell Bio Inc., Seoul, Republic of Korea			

**Table 2 materials-17-00305-t002:** Primers and sequences used in qRT-PCR.

Gene	Primer Sequence
Runt-related transcription factor 2 (*RUNX 2*)	Forward: 5′-AAG TGC GGT GCA AAC TTT CT-3′ Reverse: 5′-TCT CGG TGG CTG CTA GTG A-3
Osterix (*OSX*)	Forward: 5′-AGC CTC TGG CTA TGC AAA TGA-3′ Reverse: 5′-TGT AGA CAC TAG GCA GGC AGT CA-3
Suppressor of Mothers against Decapentaplegic (*SMAD1*)	Forward: 5′-CCA CTG GAA TGC TGT TTT CC-3′ Reverse: 5′-GTA AGC TCA TAG ACT GTC TCA AAT CC-3′
Dentin sialophosphoprotein (*DSPP*)	Forward: 5′-GGG AAT ATT GAG GGC TGG AA-3′ Reverse: 5′-TCA TTG TGA CCT GCA TCG CC-3′

## Data Availability

The datasets used/or analyzed during the current study are available from the corresponding author on reasonable request.

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
