# Peer review of "The Combined Effects on Human Dental Pulp Stem Cells of Fast-Set or Premixed Hydraulic Calcium Silicate Cements and Secretome Regarding Biocompatibility and Osteogenic Differentiation"

_materials, 2024, doi:10.3390/ma17020305_

Round 1

Reviewer 1 Report

Comments and Suggestions for Authors

Ensure that the title clearly reflects the study's main focus and outcomes. Consider specifying the type of hydraulic calcium silicate cement used if possible.

The abstract succinctly summarizes the study but could benefit from a brief mention of the methodological approach and a more explicit statement of the main conclusion to guide the reader.

Provide a more detailed background on the challenges and gaps in current knowledge that your study addresses. This will help contextualize the importance of your research.

What is the number of repetitions for the experiment? Repetitions ensure the stability and reliability of results. Also, are the human dental pulp stem cells used by the author from the same batch?

The composition and preparation process of hydraulic calcium silicate cement could be described in more detail, including the precise chemical composition ratio, particle size, and specific brand or source, to enhance the transparency and reproducibility of the experiment.

The study's key findings should be clearly elucidated in the discussion section.

Comments on the Quality of English Language

Ensure that the language is clear and professional. 

Author Response

We uploaded a PDF file.

Reviewer 2 Report

Comments and Suggestions for Authors

The manuscript, titled "The Combined Effects on Human Dental Pulp Stem Cells of Hydraulic Calcium Silicate Cement and Secretome Regarding Biocompatibility and Osteogenic Differentiation," evaluates the biocompatibility and osteogenic differentiation of HCSCs combined with secretome on human dental pulp stem cells.

The paper is interesting and written with good scientific soundness; only minor changes should be addressed.

Abstract: Describe all abbreviations when used for the first time, e.g., CCK-8....

Keywords: Should be in alphabetical order, and KEYWORDS should not contain the same words that are within the title of the text. Thus, these should be changed appropriately.

Introduction: This paragraph is a bit short. Write a paragraph describing the possible use of MTA in endodontic treatment, e.g., pulp capping, pulpotomy, perforation repair, specification, scaffold for tissue regeneration....

Author Response

We attached a PDF file below.

Reviewer 3 Report

Comments and Suggestions for Authors

I want to express my gratitude for the opportunity to review your manuscript. It is indeed a worthy project that demonstrates the potential enhancement of biocompatibility and osteogenic differentiation of HCSCs when combined with secretome, indicating a promising approach for regenerative endodontic procedures for immature permanent teeth. However, I have identified some points that require clarification and revision, as outlined below:

Minor Issues:

1. The introduction section appears to be too shallow. I kindly request that you consider expanding it to provide a more comprehensive overview of the background and context of the study.

2. There are incomplete figure captions and labeling on Figures 3 and 6. Please ensure that all figures have clear and complete captions and labeling to aid in understanding.

Major Issues:

1. It would be beneficial to elaborate on the importance of the material's surface composition on the biocompatibility results. This could provide valuable insights into the underlying factors influencing the outcomes of the study.

2. Do you believe that the composition provided by the manufacturer is consistent with the actual surface composition? It would be valuable to address any potential discrepancies or variations in the material's composition as provided by the manufacturer and its actual surface properties.

Author Response

We attached a PDF file below.
